# Efficient Partial Monitoring with Prior Information

**Hastagiri P Vanchinathan**
Dept. of Computer Science
ETH Zürich, Switzerland
hastagiri@inf.ethz.ch

**Gábor Bartók**
Dept. of Computer Science
ETH Zürich, Switzerland
bartok@inf.ethz.ch

**Andreas Krause**
Dept. of Computer Science
ETH Zürich, Switzerland
krausea@ethz.ch

## Abstract

Partial monitoring is a general model for online learning with limited feedback: a learner chooses actions in a sequential manner while an opponent chooses outcomes. In every round, the learner suffers some loss and receives some feedback based on the action and the outcome. The goal of the learner is to minimize her cumulative loss. Applications range from dynamic pricing to label-efficient prediction to dueling bandits. In this paper, we assume that we are given some prior information about the distribution based on which the opponent generates the outcomes. We propose BPM, a family of new efficient algorithms whose core is to track the outcome distribution with an ellipsoid centered around the estimated distribution. We show that our algorithm provably enjoys near-optimal regret rate for locally observable partial-monitoring problems against stochastic opponents. As demonstrated with experiments on synthetic as well as real-world data, the algorithm outperforms previous approaches, even for very uninformed priors, with an order of magnitude smaller regret and lower running time.

## 1 Introduction

We consider *Partial Monitoring*, a repeated game where in every time step a learner chooses an action while, simultaneously, an opponent chooses an outcome. Then the player receives a loss based on the action and outcome chosen. The learner also receives some feedback based on which she can make better decisions in subsequent time steps. The goal of the learner is to minimize her cumulative loss over some time horizon.

The performance of the learner is measured by the *regret*, the excess cumulative loss of the learner compared to that of the best fixed constant action. If the regret scales linearly with the time horizon, it means that the learner does not approach the performance of the best action, that is, the learner fails to learn the problem. On the other hand, sublinear regret indicates that the disadvantage of the learner compared to the best fixed strategy fades with time.

Games in which the learner receives the outcome as feedback after every time step are called *online learning with full information*. This special case of partial monitoring has been addressed by (among others) Vovk [1] and Littlestone and Warmuth [2], who designed the randomized algorithm Exponentially Weighted Averages (EWA) as a learner strategy. This algorithm achieves $\Theta(\sqrt{T \log N})$ expected regret against any opponent, where $N$ is the number of actions and $T$ is the time horizon. This regret growth rate is also proven to be optimal.

Another well-studied special case is the so-called *multi-armed bandit problem*. In this feedback model, the learner gets to observe the loss she suffered in every time step. That is, the learner does not receive any information about losses of actions she did not choose. Asymptotically optimal results were obtained by Audibert and Bubeck [3], who designed the Implicitly Normalized Forecaster (INF) that achieves the minimax optimal $\Theta(\sqrt{TN})$ regret growth rate.[1]

However, not all online learning problems have one of the above feedback structures. An important example for a problem that does not fit in either full-information or bandit problems is *dynamic pricing*. Consider the problem of a vendor wanting to sell his products to customers for the best possible price. When a customer comes in, she (secretly) decides on a maximum price she is willing to buy his product for, while the vendor has to set a price without knowing the customer's preferences. The loss of the vendor is some preset constant if the customer did not buy the product, and an "opportunity loss", when the product was sold cheaper than the customer's maximum. The feedback, on the other hand, is merely an indicator whether the transaction happened or not.

Dynamic pricing is just one of the practical applications of partial monitoring. *Label efficient prediction*, in its simplest form, has three actions: the first two actions are guesses of a binary outcome but provide no information, while the third action provides information about the outcome for some unit loss as the price. This can be thought of an abstract form of *spam filtering*: the first two actions correspond to putting an email to the inbox and the spam folder, the third action corresponds to asking the user if the email is spam or not. Another problem that can be cast as partial monitoring is that of *dueling bandits* [5, 6] in which the learner chooses a pair of actions in every time step, the loss she suffers is the average loss of the two actions, and the feedback is which action was "better".

In online learning, we distinguish different models of how the opponent generates the outcomes. In the mildest version called *stochastic* or *stationary memoryless*, the opponent chooses an outcome distribution before the game starts and then selects outcomes in an iid random manner drawn from the chosen distribution. The *oblivious adversarial* opponent chooses the outcomes arbitrarily, but without observing the actions of the learner. This selection method is equivalent to choosing an outcome sequence ahead of time. Finally, the *non-oblivious* or *adaptive adversarial* opponent chooses outcomes arbitrarily with the possibility of looking at past actions of the learner. In this work, we focus on strategies against stochastic opponents.

**Related work**   Partial monitoring was first addressed in the seminal paper of Piccolboni and Schindelhauer [7], who designed and analyzed the algorithm FeedExp3. The algorithm's main idea is to maintain an unbiased estimate for the loss of each action in every time step, and then use these estimates to run the full-information algorithm (EWA). Piccolboni and Schindelhauer [7] proved an $O(T^{3/4})$ upper bound on the regret (not taking into account the number of actions) for games for which learning is at all possible. This bound was later improved by Cesa-Bianchi et al. [8] to $O(T^{2/3})$, who also constructed an example of a problem for which this bound is optimal.

From the above bounds it can be seen that not all partial-monitoring problems have the same level of difficulty: while bandit problems enjoy an $O(\sqrt{T})$ regret rate, some partial-monitoring problems have $\Omega(T^{2/3})$ regret. To this end, Bartók et al. [9] showed that partial-monitoring problems with finitely many actions and outcomes can be classified into four groups: *trivial* with zero regret, *easy* with $\widetilde{\Theta}(\sqrt{T})$ regret, *hard* with $\Theta(T^{2/3})$ regret, and *hopeless* with linear regret. The distinguishing feature between easy and hard problems is the *local observability condition*, an algebraic condition on the feedback structure that can be efficiently verified for any problem. Bartók et al. [9] showed the above classification against stochastic opponents with the help of algorithm BALATON. This algorithm keeps track of estimates of the loss difference of "neighboring" action pairs and eliminates actions that are highly likely to be suboptimal.

Since then, several algorithms have been proposed that achieve the $\widetilde{O}(\sqrt{T})$ regret bound for easy games [10, 11]. All these algorithms rely on the core idea of estimating the expected loss difference between pairs of actions.

**Our contributions**   In this paper, we introduce BPM (Bayes-update Partial Monitoring), a new family of algorithms against iid stochastic opponents that rely on a novel way of the usage of past observations. Our algorithms maintain a confidence ellipsoid in the space of outcome distributions, and update the ellipsoid based on observations following a Bayes-like update. Our approach enjoys better empirical performance and lower computational overhead; another crucial advantage is that we can incorporate prior information about the outcome distribution by means of an initial confidence ellipsoid. We prove near-optimal minimax expected regret bounds for our algorithm, and demonstrate its effectiveness on several partial monitoring problems on synthetic and real data.

## 2 Problem setup

Partial monitoring is a repeated game where in every round, a learner chooses an action while the opponent chooses an outcome from some finite action and outcome sets. Then, the learner observes a feedback signal (from some given set of symbols) and suffers some loss, both of which are deterministic functions of the action and outcome chosen. In this paper we assume that the opponent chooses the outcomes in an iid stochastic manner. The goal of the learner is to minimize her cumulative loss.

The following definitions and concepts are mostly taken from Bartók et al. [9]. An instance of partial monitoring is defined by the *loss matrix* $L \in \mathbb{R}^{N \times M}$ and the *feedback table* $H \in \Sigma^{N \times M}$, where $N$ and $M$ are the cardinality of the action set and the outcome set, respectively, while $\Sigma$ is some alphabet of symbols. That is, if learner chooses action $i$ while the outcome is $j$, the loss suffered by the learner is $L[i,j]$, and the feedback received is $H[i,j]$.

For an action $1 \leq i \leq N$, let $\ell_i$ denote the column vector given by the $i^{\text{th}}$ row of $L$. Let $\Delta_M$ denote the $M$-dimensional probability simplex. It is easy to see that for any $p \in \Delta_M$, if we assume that the opponent uses $p$ to draw the outcomes (that is, $p$ is the *opponent strategy*), the expected loss of action $i$ can be expressed as $\ell_i^\top p$.

We measure the performance of an algorithm with its *expected regret*, defined as the expected difference of the cumulative loss of the algorithm and that of the best fixed action in hindsight:

$$R_T = \max_{1 \leq i \leq N} \sum_{t=1}^{T} (\ell_{I_t} - \ell_i)^\top p,$$

where $T$ is some time horizon, $I_t$ ($t=1,...,T$) is the action chosen in time step $t$, and $p$ is the outcome distribution the opponent uses.

In this paper, we also assume we have some prior knowledge about the outcome distribution in the form of a *confidence ellipsoid*: we are given a distribution $p_0 \in \Delta_M$ and a symmetric positive semidefinite covariance matrix $\Sigma_0 \in \mathbb{R}^{M \times M}$ such that the true outcome distribution $p^*$ satisfies

$$\|p_0 - p^*\|_{\Sigma_0^{-1}} = \sqrt{(p_0 - p^*)^\top \Sigma_0^{-1} (p_0 - p^*)} \leq 1.$$

We use the term "confidence ellipsoid" even though our condition is not probabilistic; we do not assume that $p^*$ is drawn from a Gaussian distribution before the game starts. On the other hand, the way we track $p^*$ is derived by Bayes updates with a Gaussian conjugate prior, hence the name. We would also like to note that having the above prior knowledge is without loss of generality. For "large enough" $\Sigma_0$, the whole probability simplex is contained in the confidence ellipsoid and thus partial monitoring without any prior information reduces to our setting.

The following definition reveals how we use the loss matrix to recover the structure of a game.

**Definition 1** (Cell decomposition, Bartók et al. [9, Definition 2])**.** *For any action $1 \leq i \leq N$, let $\mathcal{C}_i$ denote the set of opponent strategies for which action $i$ is optimal:*

$$\mathcal{C}_i = \{ p \in \Delta_M : \forall 1 \leq j \leq N, (\ell_i - \ell_j)^\top p \leq 0 \}.$$

*We call the set $\mathcal{C}_i$ the* optimality cell *of action $i$. Furthermore, we call the set of optimality cells $\{\mathcal{C}_1,...,\mathcal{C}_N\}$ the* cell decomposition *of the game.*

Every cell $\mathcal{C}_i$ is a convex closed polytope, as it is defined by a linear inequality system. Normally, a cell has dimension $M-1$, which is the same as the dimensionality of the probability simplex. It might happen however, that a cell is of lower dimensionality. Another possible degeneracy is when two actions share the same cell. In this paper, for ease of presentation, we assume that these degeneracies do not appear. For an illustration of cell decomposition, see Figure 1(a).

Now that we know the regions of optimality, we can define when two actions are *neighbors*. Intuitively, two actions are neighbors if their optimality cells are neighbors in the strong sense that they not only meet in "one corner".

**Definition 2** (Neighbors, Bartók et al. [9, page 4])**.** *Two actions $i$ and $j$ are* neighbors*, if the intersection of their optimality cells $\mathcal{C}_i \cap \mathcal{C}_j$ is an $M-2$-dimensional convex polytope.*

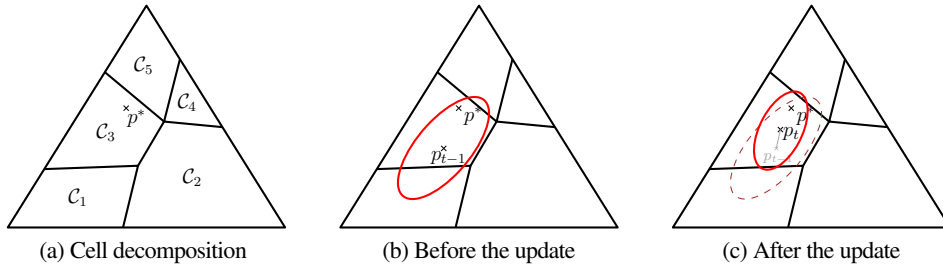

| (a) Cell decomposition | (b) Before the update | (c) After the update |

Figure 1: (a) An example for a cell decomposition with $M = 3$ outcomes. Under the true outcome distribution $p^*$, action 3 is optimal. Cells $\mathcal{C}_1$ and $\mathcal{C}_3$ are neighbors, but $\mathcal{C}_2$ and $\mathcal{C}_5$ are not. (b) The current estimate $p_{t-1}$ is far away from the true distribution, the confidence ellipsoid is large. (c) After updating, $p_t$ is closer to the truth, the confidence ellipsoid shrinks.

To optimize performance, the learner's primary goal is to find out which cell the opponent strategy lies in. Then, the learner can choose the action associated with that cell to play optimally. Since the feedback the learner receives is limited, this task of finding the optimal cell may be challenging.

The next definition enables us to utilize the feedback table $H$.

**Definition 3** (Signal matrix, Bartók et al. [9, Definition 1])**.** *Let* $\{\alpha_1, \alpha_2, ..., \alpha_{\sigma_i}\} \subseteq \Sigma$ *be the set of symbols appearing in row* $i$ *of the feedback table* $H$. *We define the* signal matrix $S_i \in \{0,1\}^{\sigma_i \times M}$ *of action* $i$ *as*

$$S_i[k,j] = \mathbb{I}(H[i,j] = \alpha_k).$$

In words, $S_i$ is the indicator table of observing symbols $\alpha_1, ..., \alpha_{\sigma_i}$ under outcomes $1, ..., M$ given that the action chosen is $i$. For an example, consider the case when the $i^{\text{th}}$ row of $H$ is $(a \quad b \quad a \quad c)$. Then,

$$S_i = \begin{pmatrix} 1 & 0 & 1 & 0 \\ 0 & 1 & 0 & 0 \\ 0 & 0 & 0 & 1 \end{pmatrix}.$$

A very useful property of the signal matrix is that if we represent outcomes with $M$-dimensional unit vectors, then $S_i$ can be used as a linear transformation to arrive at the unit-vector representation of the observation.

The following condition condition is key in distinguishing easy and hard games:

**Definition 4** (Local observability, Bartók et al. [9, Definition 3])**.** *Let actions* $i$ *and* $j$ *be neighbors. These actions are said to be* locally observable *if* $\ell_i - \ell_j \in \operatorname{Im} S_i^\top \oplus \operatorname{Im} S_j^\top$. *Furthermore, a game is* locally observable *if all of its neighboring action pairs are locally observable.*

Bartók et al. [9] showed that finite stochastic partial-monitoring problems that admit local observability have $\widetilde{\Theta}(\sqrt{T})$ minimax expected regret. In the following, we present our new algorithm family that achieves the same regret rate for locally observable games against stochastic opponents.

## 3  BPM: New algorithms for Partial Monitoring based on Bayes updates

The algorithms we propose can be decomposed into two main building blocks: the first one keeps track of a belief about the true outcome distribution and provides us with a set of *feasible* actions in every round. The second one is responsible for selecting the action to play from this action set. Pseudocode for the algorithm family is shown in Algorithm 1.

### 3.1  Update Rule

The method of updating the belief about the true outcome distribution ($p^*$) is based on the idea that we pretend that the outcomes are generated from a Gaussian distribution with covariance $\Sigma = I_M$ and unknown mean. We also pretend we have a Gaussian prior for tracking the mean. The parameters of this prior are denoted by $p_0$ (mean) and $\Sigma_0$ (covariance). In every time step, we perform a Gaussian Bayes-update using the observation received.

---

**Algorithm 1** BPM

---

    **input:** $L, H, p_0, \Sigma_0$
    **initialization:** Calculate signal matrices $S_i$
    **for** $t = 1$ **to** $T$ **do**
        Use selection rule (cf., Sec. 3.2) to choose an action $I_t$
        Observe feedback $Y_t$
        Update posterior: $\Sigma_t^{-1} = \Sigma_{t-1}^{-1} + P_{I_t}$ and $p_t = \Sigma_t \big( \Sigma_{t-1}^{-1} p_{t-1} + S_{I_t}^\top (S_{I_t} S_{I_t}^\top)^{-1} Y_t \big)$;
    **end for**

---

**Full-information case**    As a gentle start, we explain how the update rule would look like if we had full information about the outcome in each time step. The update in this case is identical with the standard Gaussian one-step update:

$$\Sigma_t = \Sigma_{t-1} - \Sigma_{t-1}(\Sigma_{t-1} + I)^{-1}\Sigma_{t-1} \qquad \text{or equiv.} \qquad \Sigma_t^{-1} = \Sigma_{t-1}^{-1} + I,$$

$$\mu_t = \Sigma_t \big( \Sigma_{t-1}^{-1} \mu_{t-1} + X_t \big) \qquad \text{or equiv.} \qquad \mu_t = \mu_{t-1} + \Sigma_t(X_t - \mu_{t-1}).$$

Here we use subindex $t-1$ for the prior parameters and $t$ for the posterior parameters in time step $t$, and denote by $X_t$ the outcome (observed in this case), encoded by an $M$-dimensional unit vector.

**General case**    Moving away from the full-information case, we face the problem of not observing the outcome, only some symbol that is governed by the signal matrix of the action we chose and the outcome itself. If we denote, as above, the outcome at time step $t$ by an $M$-dimensional unit vector $X_t$, then the observation symbol can be thought of as a unit vector given by $Y_t = S_i X_t$, provided the chosen action is $i$. It follows that what we observe is a linear transformation of the sample from the outcome distribution.

Following the Bayes update rule and assuming we chose action $i$ at time step $t$, we derive the corresponding Gaussian posterior given that the likelihood of the observation is $\pi(Y|p) \sim N(S_i p, S_i S_i^\top)$. After some algebraic manipulations we get that the posterior distribution is Gaussian with covariance $\Sigma_t = (\Sigma_{t-1}^{-1} + P_i)^{-1}$ and mean $p_t = \Sigma_t \big( \Sigma_{t-1}^{-1} p_{t-1} + P_i X_t \big)$, where $P_i = S_i^\top (S_i S_i^\top)^{-1} S_i$ is the orthogonal projection to the image space of $S_i^\top$. Note that even though $X_t$ is not observed, the update can be performed, since $P_i X_i = S_i^\top (S_i S_i^\top)^{-1} S_i X_t = S_i^\top (S_i S_i^\top)^{-1} Y_t$.

A significant advantage of this method of tracking the outcome distribution as opposed to keeping track of loss difference estimates (as done in previous works), is that feedback from one action can provide information about losses across all the actions. We believe that this property has a major role in the empirical performance improvement over existing methods.

An important part in analyzing our algorithm is to show that, despite the fact that the outcome distribution is not Gaussian, the update tracks the true outcome distribution well. For an illustration of tracking the true outcome distribution with the above update, see Figures 1(b) and 1(c).

### 3.2   Selection rules

For selecting actions given the posterior parameters, we propose two versions for the selection rule:

1. Draw a random sample $p$ from the distribution $N(p_{t-1}, \Sigma_{t-1})$, project the sample to the probability simplex, then choose the action that minimizes the loss for outcome distribution $p$. This rule is a close relative of Thompson-sampling. We call this version of the algorithm BPM-TS.

2. Use $p_{t-1}$ and $\Sigma_{t-1}$ to build a confidence ellipsoid for $p^*$, enumerate all actions whose cells intersect with this ellipsoid, then choose the action that was chosen the fewest times so far (called BPM-LEAST).

Our experiments demonstrate the performance of both versions. We analyze version BPM-LEAST.

# 4 Analysis

We now analyze BPM-Least that uses the Gaussian updates, and considers a set of feasible actions based on the criterion that an action is feasible if its optimality cell intersects with the ellipsoid

$$\left\{ p : \|p - p_t\|_{\Sigma_t^{-1}} \leq 1 + \sqrt{\frac{1}{2} N \log MT} \right\}.$$

From these feasible actions, it picks the one that has been chosen the fewest times up to time step $t$. For this version of the algorithm, the following regret bound holds.

**Theorem 1.** *Given a locally observable partial-monitoring problem $(L, H)$ with prior information $p_0, \Sigma_0$, the algorithm* BPM-Least *achieves expected regret*

$$R_T \leq C \sqrt{T N \log(MT)},$$

*where $C$ is some problem-dependent constant.*

The above constant $C$ depends on two main factors, both of them related to the feedback structure. The first one is the sum of the smallest eigenvalues of $S_i S_i^\top$ for every action $i$. The second is related to the local observability condition. As the condition says, for every neighboring action pairs $i$ and $j$, $\ell_i - \ell_j \in \mathrm{Im} S_i^\top \oplus \mathrm{Im} S_j^\top$. This means that there exist $v_{ij}$ and $v_{ji}$ vectors such that $\ell_i - \ell_j = S_i^\top v_{ij} - S_j^\top v_{ji}$. The constant depends on the maximum 2-norm of these $v_{ij}$ vectors.

The proof of the theorem is deferred to the supplementary material. In a nutshell, the proof is divided into two main parts. First we need to show that the update rule—even though the underlying distribution is not Gaussian—serves as a good tool for tracking the true outcome distribution. After some algebraic manipulations, the problem reduces to a finding a high probability upper bound for norms of weighted sums of noise vectors. To this end, we used the martingale version of the *matrix Hoeffding* inequality [12, Theorem 1.3].

Then, we need to show that the confidence ellipsoid shrinks fast enough that if we only choose actions whose cell intersect with the ellipsoid, we do not suffer a large regret. In the core of proving this, we arrive at a term where we need to upper bound $\|\ell_i - \ell_j\|_{\Sigma_t}$, for some neighboring action pairs $(i, j)$, and we show that due to local observability and the speed at which the posterior covariance shrinks, this term can be upper bounded by roughly $1/\sqrt{t}$.

# 5 Experiments

First, we run extensive evaluations of BPM on various synthetic datasets and compare the performance against CBP [10] and FeedExp3 [7]. The datasets used in the simulated experiments are identical to the ones used by Bartók et al. [10] and thus allow us to benchmark against the current state of the art. We also provide results of BPM on a dataset that was collected by Singla and Krause [13] from real interactions with many users on the Amazon Mechanical Turk (AMT) [14] crowdsourcing platform. We present the details of the datasets used and the summarize our results and findings in this section.

## 5.1 Implementation Details

In order to implement BPM, we made the following implementation choices:

1. To use BPM-Least (see Section 3.2), we need to recover the current feasible actions. We do so by sampling multiple (10000) times from concentric Gaussian ellipsoids centred at the current mean ($p_t$) and collect feasible actions based on which cells the samples lie in. We resort to sampling for ease of implementation because otherwise we deal with the problem of finding the intersection between an ellipsoid and a simplex in $M$-dimensional space.

2. To implement BPM-TS, we draw $p$ from the distribution $N(p_{t-1}, \Sigma_{t-1})$. We then project it back to the simplex to obtain a probability distribution on the outcome space.

Primarily due to sampling, both our algorithms are computationally more efficient than the existing approaches. In particular, BPM-TS is ideally suited for real world tasks as it is several orders of magnitude faster than existing algorithms during all our experiments. In each iteration, BPM-TS only needs to draw one sample from a multivariate gaussian and does not need any cell decompositions or expensive computations to obtain high dimensional intersections.

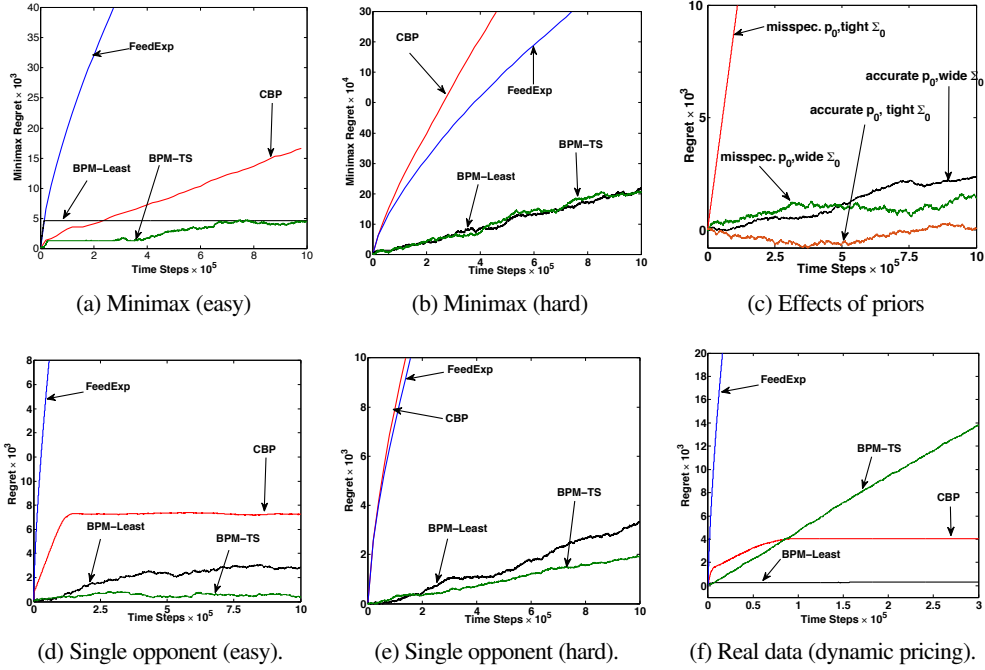

| | | |
|---|---|---|
| (a) Minimax (easy) | (b) Minimax (hard) | (c) Effects of priors |
| (d) Single opponent (easy). | (e) Single opponent (hard). | (f) Real data (dynamic pricing). |

Figure 2: (a,b,d,e) Comparing BPM on the locally non-observable game ((a,d) benign opponent; (c,e) hard opponent). Hereby, (a,b) show the pointwise maximal regret over 15 scenarios, and (d,e) show the regret against a single opponent strategy. (c) shows the effect of a misspecified prior. (f) is the performance of the algorithms on the real dynamic pricing dataset.

## 5.2 Simulated dynamic pricing games

Dynamic pricing is a classic example of partial monitoring (see the introduction), and we compare the performance of the algorithms on the small but not locally observable game described in Bartók et al. [10]. The loss matrix and feedback tables for the dynamic pricing game are given by:

$$
L = \begin{pmatrix} 0 & 1 & \cdots & N-1 \\ c & 0 & \cdots & N-2 \\ \vdots & \ddots & \ddots & \vdots \\ c & \cdots & c & 0 \end{pmatrix}; \qquad
H = \begin{pmatrix} y & y & \cdots & y \\ n & y & \cdots & y \\ \vdots & \ddots & \ddots & \vdots \\ n & \cdots & n & y \end{pmatrix}.
$$

Recall that the game models a repeated interaction of a seller with buyers in a market. Each buyer can either buy the product at the price (signal "$y$") or deny the offer (signal "$n$"). The corresponding loss to the seller is either a known constant $c$ (representing opportunity cost) or the difference between offered price and the outcome of the customer's latent valuation of the product (willingness-to-pay). A similar game models procurement processes as well. Note that this game does not satisfy local observability. While our theoretical results require this condition, in practice, if the opponent does not intentionally select harsh regions on the simplex, BPM remains applicable. Under this setting, expected *individual* regret is a reasonable measure of performance. That is, we measure the expected regret for fixed opponent strategies. We also consider the *minimax* expected regret, which measures worst-case performance (pointwise maximum) against multiple opponent strategies.

**Benign opponent** While the dynamic pricing game is not locally observable in general, certain opponent strategies are easier to compete with than others. Specifically, if the stochastic opponent chooses an outcome distribution that is away from the intersection of the cells that do not have local observability, the learning happens in "non-dangerous" or benign regions. We present results under this setting for simulated dynamic pricing with $N = M = 5$. The results shown in Figures 2(a) and 2(d) illustrate the benefits of both variants of BPM over previous approaches. We achieve an order of magnitude reduction in the regret suffered w.r.t. both the minimax and the individual regret.

**Harsh opponent** For the same problem, with opponent chooses close to the boundary of the cells of two non-locally observable actions, the problem becomes harder. Still, BPM dramatically outperforms the baselines and suffers very little regret as shown in Figures 2(b) and 2(e).

**Effect of the prior** We study the effects of a misspecified prior in Figure 2(c). As long as the initial confidence interval specified by the prior covariance is large enough to contain the opponent's distribution, an incorrectly specified prior mean does not have an adverse effect on the performance of BPM. As expected, if the prior confidence ellipse used by BPM does not contain the opponent's outcome distribution, however, the regret grows linear in time. Further, if the prior is very informative (accurately specified prior mean and tight confidence ellipse), very little regret is suffered.

## 5.3 Results on real data

**Dataset description** We simulate a procurement game based on real data. Parameter estimation was done by posting a Human Intelligence Task (HIT) on the Amazon Mechanical Turk (AMT) platform. Motivated by an application in viral marketing, users were asked about the price they would accept for (hypothetically) letting us post promotional material to their friends on a social networking site. The survey also collected features like age, geographic region, number of friends in the social network, activity levels (year of joining, time spent per day etc.). Note that since the HIT was just a survey and the questions were about a hypothetical scenario, participants had no incentives to misreport their responses. Complete responses were collected from approx. 800 participants. See [13] for more details.

**The procurement game** We simulate a procurement auction by playing back these responses offline. The game is very similar in structure to dynamic pricing, with the optimal action being the best fixed price that maximized the marketer's value or equivalently, minimized the loss. We sampled iid from the survey data and perturbed the samples slightly to simulate a stream of 300000 potential users. At each iteration, we simulate a user with a private valuation generated as a function of her attributes. We discretized the offer prices and the private valuations to be one of 11 values and set the opportunity cost of losing a user due to low pricing to be 0.5. Thus we recover a partial monitoring game with 11 actions and 11 outcomes with a 0/1 feedback matrix.

**Results** We present the results of our evaluation on this dataset in Figure 2(f). Notice that although the game is not locally observable, the outcome distribution does not seem to be in a difficult region of the cell decomposition as the adaptive algorithms (CBP and both versions of BPM) perform well. We note that the total regret suffered by BPM-LEAST is a *factor of 10 lower* than the regret achieved by CBP on this dataset. The plots are averaged over 30 runs of the competing algorithms on the stream. To the best of our knowledge, this is the first time partial monitoring has been evaluated on a real world problem of this size.

## 6 Conclusions and future work

We introduced a new family of algorithms for locally observable partial-monitoring problems against stochastic opponents. We also enriched the model of partial monitoring with the possibility of incorporating prior information about the outcome distribution in the form of a confidence ellipsoid. The new insight of our approach is that instead of tracking loss differences, we explicitly track the true outcome distribution. This approach not only eases computational overhead but also helps achieve low regret by being able to transfer information between actions. In particular, BPM-TS runs orders of magnitude faster than any existing algorithms, opening the path for the model of partial monitoring to be applied on realistic settings involving large numbers of actions and outcomes.

Future work includes extending our method for adversarial opponents. Bartók [11] already uses the idea of tracking the true outcome distribution with the help of a *confidence parallelotope*, which is rather close to our approach, but has the same shortcomings as other algorithms that track loss differences: it can not transfer information between actions. Extending our results to problems with large action and outcome spaces is also an important direction: if we have some prior information about the similarities between outcomes and/or actions, we have a chance for a reasonable regret.

**Acknowledgments** This research was supported in part by SNSF grant 200021‗137971, ERC StG 307036 and a Microsoft Research Faculty Fellowship.

## Footnotes

[1]The algorithm Exp3 due to Auer et al. [4] achieves the same rate up to a logarithmic factor.

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
