[Supplementary Material]

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

# A Proof of Theorem 1

## A.1 Validity of the update

We assume that $p^*$, the true opponent strategy, is within some distance from our initial prior $p_0$, measured in $\Sigma_0^{-1}$-distance:

$$\|p_0 - p^*\|_{\Sigma_0^{-1}} \leq 1.$$

First we observe that the update can be rewritten in a cumulative form, to see how the parameters change from the initial prior $(p_0, \Sigma_0)$:

$$\Sigma_t^{-1} = \Sigma_0^{-1} + \sum_{s=1}^{t} P_{I_s}$$

$$\Sigma_t^{-1} p_t = \Sigma_0^{-1} p_0 + \sum_{s=1}^{t} P_{I_s} X_s.$$

Now let us investigate the $\Sigma_t^{-1}$-distance of $p_t$ from $p^*$!

$$\|p_t - p^*\|_{\Sigma_t^{-1}} = \left\| \Sigma_t \Sigma_0^{-1} p_0 + \Sigma_t \sum_{s=1}^{t} P_{I_s} X_s - p^* \right\|_{\Sigma_t^{-1}}$$

Now we decompose the samples $X_s$ to mean and noise with the new notation $X_s = p^* + \epsilon_s$, yielding

$$\|p_t - p^*\|_{\Sigma_t^{-1}} = \left\| \Sigma_t \Sigma_0^{-1} p_0 + \Sigma_t \sum_{s=1}^{t} P_{I_s} (p^* + \epsilon_s) - p^* \right\|_{\Sigma_t^{-1}}$$

$$= \left\| \Sigma_t \Sigma_0^{-1} p_0 + \Sigma_t \underbrace{\left( \sum_{s=1}^{t} P_{I_s} - \Sigma_t^{-1} \right)}_{-\Sigma_0^{-1}} p^* + \Sigma_t \sum_{s=1}^{t} P_{I_s} \epsilon_s \right\|_{\Sigma_t^{-1}}$$

$$\leq \left\| \Sigma_t \Sigma_0^{-1} (p_0 - p^*) \right\|_{\Sigma_t^{-1}} + \left\| \Sigma_t \sum_{s=1}^{t} P_{I_s} \epsilon_s \right\|_{\Sigma_t^{-1}}.$$

We deal with the two resulting terms separately.

$$\left\| \Sigma_t \Sigma_0^{-1} (p_0 - p^*) \right\|_{\Sigma_t^{-1}}^2 = (p_0 - p^*)^\top \Sigma_0^{-1} \underbrace{\Sigma_t \Sigma_0^{-1}}_{(I - \Sigma_0^{-1} \sum_{s=1}^{t} P_{I_s})^{-1}} (p_0 - p^*)$$

$$\leq \|p_0 - p^*\|_{\Sigma_0^{-1}} \leq 1.$$

The second term is harder. Basically this is the term where we "pay the price" for not having started with a Gaussian distribution. We need to show that

$$\left\| \Sigma_t \sum_{s=1}^{t} P_{I_s} \epsilon_s \right\|_{\Sigma_t^{-1}} = \left\| \sum_{s=1}^{t} \sqrt{\Sigma_t} P_{I_s} \epsilon_s \right\|$$

is bounded with high probability. For any given action sequence, the above expression is a sum of independent random matrices. Now we recite a concentration inequality we need:

**Theorem 2** (Matrix Hoeffding Theorem [12, Theorem 1.3]). *Consider a finite sequence $\{X_k\}$ of independent, random, self-adjoint matrices with dimension $d$, and let $\{A_k\}$ be a sequence of fixed self adjoint matrices. Assume that each random matrix satisfies*

$$\mathbb{E} X_k = 0 \qquad \text{and} \qquad X_k^2 \preccurlyeq A_k^2 \qquad \text{almost surely.}$$

*Then, for all $t \geq 0$,*

$$P\left( \left\| \sum_k X_k \right\|_2 \geq t \right) \leq d \exp\left( -t^2 / 8\sigma^2 \right) \qquad \text{where} \qquad \sigma^2 = \left\| \sum_k A_k^2 \right\|_2.$$

The above theorem can be extended to rectangular matrices, using the "dilation trick"[2]: for rectangular matrices $B_k \in \mathbb{R}^{d_1 \times d_2}$, we use the theorem with

$$X_k = \begin{pmatrix} 0 & B_k \\ B_k^\top & 0 \end{pmatrix} \in \mathbb{R}^{d_1 + d_2}.$$

In our case, $X_s = \sqrt{\Sigma_t} P_{I_s} \epsilon_s$. Also note that here we need the martingale version of the inequality, which also holds, according to Section 7 of Tropp [12]. After algebraic manipulations, we arrive at

$$P\left(\left\|\sum_s P_s \epsilon_s\right\|_{\Sigma_t^{-1}} \geq \sqrt{\frac{1}{2}N \log \frac{M+1}{\delta}}\right) \leq \delta.$$

Putting together the terms we get that

$$\|p_t - p^*\|_{\Sigma_t^{-1}} \leq 1 + \sqrt{\frac{1}{2}N \log \frac{M+1}{\delta}}$$

with probability at least $1 - \delta$.

## A.2   Regret

Now we turn our attention to calculating the regret of the algorithm that chooses the action that is chosen the fewest times so far among the actions whose optimality cells intersect with the current confidence ellipsoid. To accommodate the error for the outcome distribution not being Gaussian, we use the ellipsoid defined as

$$\left\{p : \|p - p_t\|_{\Sigma_t^{-1}} \leq 1 + \sqrt{\frac{1}{2}N \log \frac{M+1}{\delta}}\right\}.$$

The regret in a turn results from choosing a suboptimal action. Let us assume wlog that the optimal action is action 1, the true opponent strategy is $p^*$, and the chosen action is action $k$. Then, the instantaneous regret is

$$r_t = (\ell_k - \ell_1)^\top p^*.$$

Now if we pick a point $p$ in the intersection of the cell of action $k$ and the confidence ellipse, we can connect $p^*$ and $p$ with a line segment. That segment goes through the cells of, say, $1 = i_0, i_1, ..., i_d = k$. Then we can write

$$(\ell_k - \ell_1)^\top p^* = \sum_{j=1}^d (\ell_{i_j} - \ell_{i_{j-1}})^\top p^*$$

$$= \sum_{j=1}^d (\ell_{i_j} - \ell_{i_{j-1}})^\top (p^j - p^*),$$

where we denote by $p^j$ the point where our line segment intersects the boundary of cells $i_{j-1}$ and $i_j$. The above equation is true because for every $j$, $(\ell_{i_j} - \ell_{i_{j-1}})^\top p_j = 0$. Now we upper bound, for every $j$, the term

$$(\ell_{i_j} - \ell_{i_{j-1}})^\top (p^j - p^*) \leq \|\ell_{i_j} - \ell_{i_{j-1}}\|_{\Sigma_t} \|p^j - p^*\|_{\Sigma_t^{-1}},$$

with the help of Hölder's inequality. We know from the previous section that $\|p^j - p^*\|_{\Sigma_t^{-1}}$ can be upper bounded with high probability. It remains to upper bound the first term.

With the help of the local observability condition, we have

$$\ell_{i_j} - \ell_{i_{j-1}} = S_{i_j}^\top v_{i_j, i_{j-1}} - S_{i_{j-1}}^\top v_{i_{j-1}, i_j},$$

for some $v_{i_{j-1}, i_j}, v_{i_j, i_{j-1}}$, and thus the problem reduces to upper bounding $\|S_i^\top v_{i,i'}\|_{\Sigma_t}$ for all $1 \leq i, i' \leq N$:

$$\|S_i^\top v_{i,i'}\|_{\Sigma_t}^2 = \left\|\sqrt{S_i \Sigma_t S_i^\top} v_{i,i'}\right\|_2^2$$

$$\leq \|S_i \Sigma_t S_i^\top\|_2 \|v_{i,i'}\|_2^2$$

$$\leq \|S_i \Sigma_t S_i^\top\|_2 V_{\max}^2,$$

_______________________

[2]See remark 3.11 in Tropp [12].

where $V_{\max} = \max_{1 \le i, i' \le N} \|v_{i,i'}\|_2$.

$$\left\| S_i \Sigma_t S_i^\top \right\|_2 = \left\| S_i \left( \Sigma_0^{-1} + \sum_{s=1}^{t} P_{I_s} \right)^{-1} S_i^\top \right\|_2$$

$$\le \left\| S_i \left( \Sigma_0^{-1} + n_i P_i \right)^{-1} S_i^\top \right\|_2$$

$$= \left\| (S_i^+)^+ \left( \Sigma_0^{-1} + n_i P_i \right)^{-1} (S_i^{\top +})^+ \right\|_2$$

$$\le \left\| \left( n_i S_i^{\top +} S_i^\top (S_i S_i^\top)^{-1} S_i S_i^+ \right)^+ \right\|_2$$

$$\le \frac{c_i}{n_i}$$

for some constant $c_i$.

Putting everything together we have that the instantaneous regret at time step $t$ is

$$r_t \le 2 V_{\max} K_i \sqrt{\frac{C_1}{n_{\min}}} \left( C + \sqrt{\frac{1}{2} N \log \frac{M+1}{\delta}} \right).$$

Since our algorithm picks the action that is chosen the fewest number of times, it ensures that $n_{\min} \ge t/N$. Summing up the instantaneous regret for every turn we get the desired result

$$R_T \le C_2 \sqrt{T N \log M T / \delta} \qquad\qquad \text{w.p.} \ge 1 - \delta.$$

Setting $\delta$ to $1/\sqrt{T}$, we get the desired result.