[Reviews · NeurIPS 2014]

Submitted by Assigned_Reviewer_3

The authors give an algorithm for "easy" partial-monitoring games, ones that satisfy the local observability condition of Bartok et al. Their algorithm BPM attains the O(\sqrt{T}) rate which is minimax optimal for such games.

Originality and Significance:
There are already algorithms that attain O(\sqrt{T}) regret for "easy" partial monitoring games. Indeed, the authors compare themselves against the CBP algorithm of Bartok et al. Their contribution is therefore improved empirical performance over existing algorithms, as well as the ability to incorporate a prior over the outcome distribution.

They consider only finite-outcome games (as in Bartok et al), against stochastic opponents. However, this still encompasses important settings such as the dynamic pricing application on which they run experiments.

Their algorithm is novel. Rather than estimating the loss difference between each pair of action, they update their posterior over the outcome distribution. The trick relies on the observation that their "signal matrices" S_i give a linear transformation between the algorithm's observation on each round and the underlying realized outcome.

Clarity:
The paper is very clear, well-written, and easy to read.

Quality:
The theoretical results are easy to follow and appear correct. They also run convincing experiments in dynamic pricing. The first set of results show improved results over previous methods (CBP, FeedExp) in simulations. However, they go further to collect real pricing data gathered from mechanical turk surveys, running experiments that simulate users with those valuations. Their analyzed algorithm -- BPM-LEAST continues to outperform all methods. It's interesting that the Thompson sampling approach seems to suffer linear regret in this experiment: any idea why?
Summary: The authors give an algorithm for "easy" partial-monitoring games with stochastic outcomes, attaining the minimax rate of O(\sqrt{T}). The approach is novel and the authors demonstrate clear empirical advantages over previous such algorithms.

Submitted by Assigned_Reviewer_5

Summary
=============

The paper proposes a novel algorithm for the problem of online learning with partial monitoring against a stochastic adversary. The proposed algorithm (BPM) can incorporate a prior in the form of a Gaussian distribution over the parameter defining the outcome distribution. This Gaussian assumption is only used to define the algorithm, while its regret performance is analyzed for any possible outcome distribution. The regret analysis shows that BPM achieves similar performance as existing algorithms, while an empirical study on synthetic and real data shows a significant advantage both computationally and in terms of the regret.

Quality
==============

The paper is of very good quality in general.

1- The algorithm is very simple, since it boils down in computing a Gaussian posterior over a linear model and the selection rules are clear and well motivated as well. The only aspect which is not discussed in a rigorous way its the computational cost of the algorithm compared to existing algorithms. In fact, BPM still requires to invert a MxM matrix at each step and the selection rule BPM-LEAST requires a random sampling since its actual exact implementation is computationally very heavy.
2- Although the theoretical analysis matches existing results, it is useful in assessing the soundness of BPM (at least in its BPM-LEAST variant). There are only a couple of aspects that I would ask the authors to comment in more detail:
a- The regret is studied only in the case of locally observable games. What is the regret in case a non locally observable games? Is it possible to achieve the T^{2/3} regret for "hard" problems? This part is covered by the experiments but a theoretical comment would be useful.
b- In the statement of Thm. 1, C is a problem-dependent quantity. It would useful to make explicit the characteristics of the problem contributing to C.
c- The Thm. 2 used in the proof is defined for any *fixed* sequence of matrices. In the case of BPM the quantity X_s is random with dependence on the actions selected in the past. It seems like this may pose a problem since the samples are not iid anymore (which would require an Azuma version of the statement) and additional dependencies on the dimensionality of X_s may appear (in the same flavor as in the Lemma in

Yasin Abbasi-Yadkori, Dávid Pál, and Csaba Szepesvári. Improved algorithms for linear stochastic bandits. In Advances in Neural Information Processing Systems (NIPS), 2011.

It'd be great if the authors could explain whether this is the case or whether Thm.2 is indeed enough.

3- In the experiments the authors consider non locally observable problems and show a significant improvement w.r.t. the state of the art with a much smaller regret in all the synthetic data considered in Section 5.2. The other interesting aspect is that a pseudo-real-data experiment is run. I call it "pseudo" since data are real but the experiment is still generated artificially from the data. Nonetheless, I think it is a significant result. The only thing which is really surprising is the difference in performance between the two variants of PBM. What is your explanation for it?

Clarity
==============

The paper is well written and clear.

Originality
==============

The idea used in the paper is not very original since it reuses tools available in the literature. Nonetheless, the result is novel enough.

Significance
==============

Overall the result is significant since it provides an efficient and easy algorithm to solve the challenging problem of partial monitoring in online learning. Preliminary results including real data support the fact that the algorithm may find application in real problems.
Summary: My evaluation of the paper is positive. The algorithm is simple but principled. The theoretical analysis matches existing bounds but confirms that the algorithm is sound. In practice, the algorithm is more efficient than the state-of-the-art and its empirical performance is clearly better.

Submitted by Assigned_Reviewer_28

This paper considers the well-studied partial monitoring problem under the local observability condition against a stochastic adversary.The additional twist is that the learner is provided an ellipsoid in the space of probability distributions over the adversary's moves that contains the adversary's strategy. The paper gives an online learning algorithm that, starting from such an ellipsoid, successively refines it based on evidence gathered during the run of the algorithm and chooses actions from the intersection of the natural cell decomposition of the the space of probability distributions with the current ellipsoid. The authors prove that the algorithm enjoys an optimal O(sqrt{T}) regret bound, matching known regret bounds.

The chief advantage of the algorithm is that it is very efficient to implement compared to the previous algorithms (although the authors do not provide a precise comparison). Furthermore, experiments show that the regret achieved by the algorithm is significantly lower than previously known algorithms, even in situations where local observability is not necessarily satisfied. One thing that is lacking from the paper is a proper comparison of the regret bounds achieved in this paper to previous ones.
Summary: This is a well-written paper which applies a novel idea of refining confidence ellipsoids to the problem of partial monitoring under the local observability condition, and gives an algorithm with an optimal regret bound which is more efficient than previous approaches and great experimental performance. Overall, a significant result.
Author Feedback
Author rebuttal: We thank the reviewers for they thorough and insightful reviews. Below we address all the issues raised by the reviewers.

Computational complexity: We agree with the comment that a thorough comparison of the algorithms in terms of computational complexity is missing. We will include it in the next version.

Thompson sampling with linear regret: First of all, we do not have an analysis for the TS version of the algorithm yet, so it might just be the case that it does not work. Another explanation could be that while the algorithm has sublinear regret, the transitional phase of the algorithm (the number of time steps before it starts to learn) is higher than usual. Also note that we tried the TS version on a not locally observable game, although with a prior that makes sure that the outcome distribution is in an area of the probability simplex where local observability is satisfied.

Not locally observable games: While the present algorithm enjoys provable \sqrt{T} regret on locally observable instances, in general the algorithm can have linear regret if the game it is run on is not locally observable. However, if the true outcome distribution is in an area of the probability simplex where local observability holds, we observe that the algorithm behaves as if the game was locally observable. For an adaptive algorithm (an algorithm that achieves T^{2/3} regret for non locally observable games), we probably need an extension of the algorithm similar in taste to that of Bartok et al. (ICML 2012).

Problem dependent constant in the bound: We will include a more thorough description on how the problem at hand influences the constant.

Concentration in Thm 2: The reviewer is right! Thank you for noticing this bug. Indeed, we need the martingale version of the concentration bound used in our proof. In the paper we cited for the matrix-Hoeffding inequality [12], section 7 contains a martingale version of the bound we need, without any difference in the bound itself.

Significant improvement compared to previous algorithms: We believe that the main reason for such a surprising improvement is that the new algorithm can connect information coming from different actions. That is, choosing action 1 gives us information about the loss differences of action pairs not containing action 1. While the analysis does not show this difference, in practice the improvement is significant.